# Molecular Mechanisms of Cartilage Repair and Their Possible Clinical Uses: A Review of Recent Developments

**DOI:** 10.3390/ijms232214272

**Published:** 2022-11-17

**Authors:** Emérito Carlos Rodríguez-Merchán

**Affiliations:** 1Department of Orthopedic Surgery, La Paz University Hospital, 28046 Madrid, Spain; ecrmerchan@hotmail.com; 2Osteoarticular Surgery Research, Institute for Health Research, La Paz University Hospital, Autonomous University of Madrid, 28046 Madrid, Spain

**Keywords:** cartilage, joint, repair, molecular mechanisms, clinical applications

## Abstract

Articular cartilage (AC) defects are frequent but hard to manage. Osteoarthritis (OA) is a musculoskeletal illness that afflicts between 250 and 500 million people in the world. Even though traditional OA drugs can partly alleviate pain, these drugs cannot entirely cure OA. Since cartilaginous tissue of the joints has a poor self-repair capacity and very poor proliferative ability, the healing of injured cartilaginous tissue of the joint has not been accomplished so far. Consequently, the discovery of efficacious mediations and regenerative treatments for OA is needed. This manuscript reviews the basic concepts and the recent developments on the molecular mechanisms of cartilage repair and their potential clinical applications. For this purpose, a literature exploration was carried out in PubMed for the years 2020, 2021, and 2022. On 31 October 2022 and using “cartilage repair molecular mechanisms” as keywords, 41 articles were found in 2020, 42 in 2021, and 36 in 2022. Of the total of 119 articles, 80 were excluded as they were not directly related to the title of this manuscript. Of particular note are the advances concerning the mechanisms of action of hyaluronic acid, mesenchymal stem cells (MSCs), nanotechnology, enhancer of zeste 2 polycomb repressive complex 2 subunit (EHZ2), hesperetin, high mobility group box 2 (HMGB2), α2-macroglobulin (α2M), proteoglycan 4 (Prg4)/lubricin, and peptides related to cartilage repair and treatment of OA. Despite the progress made, current science has not yet achieved a definitive solution for healing AC lesions or repairing cartilage in the case of OA. Therefore, further research into the molecular mechanisms of AC damage is needed in the coming decades.

## 1. Introduction

Articular cartilage (AC) is a compact extracellular matrix (ECM)-rich tissue that degenerates due to continuing mechanical stress, leading to osteoarthritis (OA). The tissue has poor intrinsic repair capacity particularly in elderly and osteoarthritic articulations. OA is the most frequent musculoskeletal illness, impacting between 250 and 500 million people in the world [1,2].

AC is an avascular tissue. Consequently, diffusivity is an essential carrier instrument for nutrients and other molecular signals controlling its cell metabolism and preservation of the ECM. Comprehending how solutes distribute into joint cartilaginous tissue is paramount to clarifying its disorders, and to creating approaches for repair and reparation of its ECM [3].

Cells in AC are zonal organized. Cells in the external area of AC are usually little in size and proliferative. Stimulation with adequate negative pressure is advantageous to cell endurance and tissue repair [4]. AC defects are frequent in the clinic but hard to manage [5]. AC degradation has been identified as the principal abnormal finding in OA. The processes that control the change from normal cartilage to cartilage with OA endure undefined [6]. OA is the most frequent chronic and degenerative articular illness. Even though traditional OA drugs can partly alleviate pain, these drugs cannot entirely cure OA [7]. OA is the most frequent articular illness, lacking illness-modifying therapies [8].

The ECM of AC is a system with three dimensions (3D) principally composed of entangled fibrils of collagen and aggregates of interfibrillar aggrecan. Throughout the appearance of OA, the ECM is put through a compound of chemical and structural modifications that are essential in the start and the advancement of the illness. Although the molecular processes implicated in the pathological remodeling of the ECM are deemed as critical, they remain, nevertheless, not entirely clarified [9].

Due to the fast aging of the world population, the amount of people with articular illness continues to rise. Consequently, a great number of subjects need adequate management for OA. Additionally, OA can substantially affect the subject’s quality of life (QoL) due to knee joint discomfort, muscle weakness in the lower limbs, or ambulatory problems. Due to the fact that AC has a poor self-repair capacity and a very poor low proliferative ability, healing of injured AC has not been accomplished so far. Contemporary pharmacological management of OA is restricted to the negligible relief of pain (e.g., intra-articular injection of hyaluronic acid or nonsteroidal anti-inflammatory medications); therefore, the discovery of efficacious medications and regenerative treatments for OA is greatly required [10].

The risk circumstances for the appearance of OA and the contemporary pharmacological and nonpharmacological approaches for its management are shown in Figure 1 [11].

The repair of AC defects is still defying orthopedic surgery due to the avascular configuration of AC and the restricted regenerative ability of adult chondrocytes [12].

For this purpose, a literature exploration was carried out in PubMed for the years 2020, 2021, and 2022. On 31 October 2022 and using “cartilage repair molecular mechanisms” as keywords, 41 articles were found in 2020, 42 in 2021, and 36 in 2022. Of the total of 119 articles, 80 were excluded as they were not directly connected to the title of this manuscript. Therefore, 39 articles were finally analyzed, the results of which are presented below.

## 2. Basic Concepts

AC has a restricted ability for repair. The first try to repair cartilaginous tissue utilizing tissue engineering was published in 1977. Since then, cell-based procedures have been used in orthopedic surgery [13]. There are a number of elements implicated in cellular treatment and tissue engineering for the repair of cartilaginous tissue. Treatments for the repair of cartilaginous tissue need any compound of cells, biomaterials, mechanical loading, and/or bioactive effectors (Figure 2) [13].

Figure 3 shows a schematic summary of approaches for the repair of cartilaginous tissue [13]. All aforementioned treatments can be improved with bioactive effectors or mechanical loading.

Figure 4 shows the main emergent therapeutic goals in OA. Pathological goals mainly gather into those facilitating repair, those neutralizing pain, and those constraining tissue swelling. There is a mutual association between inflammatory and repair processes in the OA articulation, both of which impact pain [14].

Figure 5 shows an outline of nanotechnology-based uses in osteoarthritis (OA) diagnostics and treatment [11].

## 3. Recent Developments

### 3.1. Hyaluronic Acid

In a murine study, Luo et al. discovered that intra-articular injections of hyaluronic acid facilitated the chondrogenic differentiation of human amniotic mesenchymal stem cells (hAMSCs). In other words, hyaluronic acid notably improved the expression of chondrocyte-specific markers including collagen type 2 alpha 1 (Col2α1), acan, and SRY-box transcription factor 9 (Sox9) in hAMSCs, with intense synergistic impact on chondrogenic differentiation, together with the frequently utilized inducer, transforming growth factor β3 (TGF-β3). Hyaluronic acid turns on the RASL11B gene to stimulate the chondrogenic differentiation of hAMSCs via the activation of Sox9 and extracellular signal-regulated protein kinase (ERK)/Smad signaling, therefore rendering an approach for the repair of cartilaginous defects by hyaluronic acid-based stem cell treatment [15]. Galla et al. published a new form of hyaluronic acid that seemed to be well-absorbed and disseminated to chondrocytes, maintaining their biological functions. Hence, the oral administration of GreenIuronic^®^ (Vivatis Pharma GBHE, Grüner Deich 1-3, Hamburg, Germany) in humans can be deemed a sound approach to attain an advantageous therapeutic impact on OA [16].

### 3.2. Mesenchymal Stem Cells (MSCs)

Intra-articular injections of MSCs have been examined as a possible treatment for the management of knee OA, with some proof of accomplishment in initial human trials. Salerno et al. showed that ample in vitro aging of bone marrow-derived human MSCs causes a decline of chondrogenesis but no decrease in trophic repair. By combining transcriptomic and proteomic information utilizing Ingenuity pathway analysis, Salerno et al. discovered that diminished chondrogenesis with passage is connected to downregulation of the forkhead transcription factor M1 (FOXM1) signaling pathway while preservation of trophic repair is connected to C-X-C motif chemokine ligand 12 (CXCL12). In a try at creating functional markers of MSC power, Salerno et al. recognized the decline of messenger ribonucleic acid (mRNA) expression for metalloproteinase 13 (MMP13) was connected with the decline of chondrogenic power of MSCs and the endured release of elevated amounts of TIMP metallopeptidase inhibitor 1 (TIMP1) protein was connected with the preservation of trophic repair ability. Salerno et al. reckoned that early passage MMP13^+^, TIMP1-secreting^high^ MSCs should be utilized for autologous OA treatments devised to perform through engraftment and chondrogenesis, while later passage MMP13^−^, TIMP1-secreting^high^ MSCs could be used for allogeneic OA treatment devised to perform through trophic repair [17].

Zhang et al. observed that miR-199b-5p overexpression restrained the growth of C3H10T1/2 cells but facilitated transforming growth factor-β3 (TGF-β3)-induced C3H10T1/2 cells of chondrogenic differentiation. The findings of this study might render a new perception on miRNA-mediated MSC treatment for cartilage-related conditions [18].

Casari et al. stated that adipose tissue (AT) has turned into a fountain of MSCs for regenerative medicine employments, particularly skeletal diseases. They compared AT processed by centrifugation (C-AT) with microfragmentation (MF-AT). Bringing into sharp focus MF-AT, Casari et al. afterward evaluated the effect of synovial fluid (SF) alone on both MF-AT and isolated AT-MSC to better comprehend their cartilage repair processes. The findings of this study showed that MF has a positive impact on the preservation of AT histology and might activate the expression of trophic factors that ameliorate tissue repair by processed AT [19].

According to Ren et al., the stemness and differentiation traits of bone marrow mesenchymal stem cells (BMSCs) in three-dimensional (3D) culture are very important for stem cell treatment and engineering repair of cartilaginous tissue [20].

Yan X et al. showed that MSC treatment using a deoxyribonucleic acid (DNA) supramolecular hydrogel facilitated the creation of quality cartilage, diminished bone spurs, and normalized subchondral osseous tissue under the elevated friction condition of OA [2].

A study from Yan B et al. established the anti-OA effectiveness, safety, and a paracrine-based mechanism of adipose-derived stem cells (ADSCs), rendering an encouraging cell-based treatment alternative for OA management [8].

Norouzi-Barough et al. reported the therapeutic possibility of MSC-derived exosomes as a cell-free therapy method for the management of cartilage defects [21].

Feng et al. reported that 3N-cadherin can eventually facilitate chondrogenic differentiation of BMSCs by restraining the Wnt signaling pathway. They compared and examined the impact of N-cadherin on chondrogenic differentiation of BMSCs to analyze the related mechanism, so as to render a theoretical basis for the clinical treatment (repair and regeneration) of AC injuries [22].

According to Zhang et al., induced pluripotent stem cells (iPSCs) are a limitless fountain for cartilage regeneration as they can produce a broad range of cell types. They encountered that in chondrogenic induction medium, shaking culture alone substantially upregulated the chondrogenic markers SOX9, Col2a1, and aggrecan in iPSCs-Tet/BMP-4 by day 21. The culture system reported by Zhang et al. could be a valuable implement for further research of the mechanism of BMP-4 in controlling iPSC differentiation toward the chondrogenic lineage and should ease investigation in cartilage development, repair, and OA [23].

### 3.3. Mechanisms of Diffusion in Articular Cartilage (AC)

Travascio et al. observed that diffusivity was impacted by molecular size, with the extent of the diffusion coefficients diminishing as the Stokes radius of the probe augmented. This report rendered new data on the mechanisms of diffusion in AC. The results of this study can be exploited to further analyze OA and to plan therapies for the restoration or replacement of cartilaginous tissue [3].

### 3.4. Mechanisms of miRNA Control of the Transcriptome of Tissue-Engineered Cartilage in Reaction to IL-1β and TNF-α

In an in vitro murine-induced pluripotent stem cell (miPSC) experiment, Ross et al. demonstrated that delivery of miR-17-5p and miR-20a-5p imitates substantially diminished degradative enzyme activity levels while also reducing expression of swelling-related genes in cytokine-treated cells [24].

### 3.5. SIRT1 has an Essential Coordination Function in BMP2-Induced Chondrogenic Differentiation of Stem Cells and Cartilage Preservation under Oxidative Stress

Lu et al. affirmed that silent mating type information regulator 2 homolog-1 (SIRT1) is an essential histone deacetylase that controls proliferation, differentiation, aging, and swelling processes; moreover, it is a crucial factor for chondrogenesis. In their study, Lu et al. established the experimental basis for examining the utilization of SIRT1 in the repair of cartilage defects [5].

### 3.6. Nanotechnology for OA Clinical Management

Mohammadinejad et al. stated that nanotechnology can be beneficial for the diagnosis, follow-up, and treatment of OA (Figure 5). Nanotubes, magnetic nanoparticles, and other nanotechnology-based medication and gene delivery systems may be utilized for aiming molecular pathways and pathogenic processes implicated in OA appearance [11]. According to Li et al., bioactive extracellular vesicle (EV)-based nanotherapeutics have revealed new views for clinicians, making feasible strong instruments and therapies for regenerative medicine (cartilage repair and regeneration) [25].

### 3.7. Ezh2 Ameliorates Osteoarthritis (OA) by Triggering TNFSF13B

Du et al. examined the expression of EZH2 (enhancer of zeste 2 polycomb repressive complex 2 subunit), an H3K27me3 (histone H3 lysine 27 trimethylation) transferase, in human OA cartilages and its regulations in controlling OA pathogenesis. The results of this study revealed that an EZH2-positive subpopulation existed in OA subjects and that EZH2-TNFSF13B signaling was accountable for controlling chondrocyte healing and hypertrophy. Therefore, EZH2 may act as a new possible objective for the diagnosis and management of OA [26].

### 3.8. Hesperetin (HPT) Averts Cartilage Degradation

Wu et al. studied whether hesperetin (HPT) had chondroprotective impact against the tumor necrosis factor-α (TNF-α)-induced inflammatory reaction of chondrocytes and connected mechanisms and determined the effect of HPT on OA induced by anterior cruciate ligament transection (ACLT). The results of this study demonstrated that HPT has a substantial protective and anti-inflammatory impact on chondrocytes via the AMPK signaling pathway, efficaciously averting cartilage degeneration. Taking into account the diverse beneficial impacts of HPT, it can be utilized as a possible natural medication to manage OA [27].

### 3.9. Intrinsic Restoration Reaction in Cartilage, Mediated by Aggrecan-Dependent Sodium Flux

Keppie et al. reported an intrinsic reparation mechanism, regulated by matrix stiffness and mediated by the free sodium concentration, in which heparan sulfate-bound growth factors are liberated from cartilage following detrimental load. They recognized aggrecan as a depot for sequestered sodium, explicating why OA tissue loses its capacity to repair. Medications that re-establish matrix sodium to permit adequate liberation of growth factors following load were foreseen to facilitate intrinsic cartilage repair in OA [1].

### 3.10. Clock Knockdown Attenuated Reactive Oxygen Species-Mediated Senescence of Chondrocytes via Re-Establishing Autophagic Flux

Zhong et al. studied the possible function and mechanism of the circadian gene clock in OA pathology. They encountered that clock knockdown can attenuate ROS-mediated senescence of chondrocytes via re-establishing autophagic flux in noncircadian mode, rendering a possible therapeutic goal for OA [6].

### 3.11. HMGB2 Facilitates Chondrocyte Proliferation under Negative Pressure via the Phosphorylation of AKT

Liu et al. found that high-mobility group box 2 (HMGB2) exhibited an improvement effect on chondrocyte proliferation under negative pressure through the phosphorylation of AKT [4].

### 3.12. Regenerative Potential of OA Human Infrapatellar Fat Pad (IPFP)-Derived Progenitors Osteochondral Defects

In an animal experiment performed in femoral bones of mice, van Schaik et al. demonstrated viability in producing ex vivo osteochondral defects and showed the regenerative capacity of OA human IPFP-derived progenitors. Therefore, the rat model reported by van Schaik et al. could be utilized to investigate the impact of aging and OA on tissue regeneration and to examine molecular processes of cartilage repair utilizing genetically modified mice [28].

### 3.13. Function and Mechanism of Arginine–Glycine–Aspartic Acid (Arg–Gly–Asp, RGD) Peptide Family in Cartilage Tissue Engineering

According to Yang et al., given that the majority of the scaffold materials utilized in cartilage tissue engineering are biologically inactive, it is crucial to augment the cellular adhesion capacity throughout tissue engineering restoration. RGD polypeptide families were deemed as appropriate prospects for the management of a diversity of illnesses and for the regeneration of diverse tissues [29].

### 3.14. 3D Bioprinting Gradient-Structured Constructs for Anisotropic Cartilage Regeneration

Sun et al. created an anisotropic gradient-structured cartilage scaffold by 3D bioprinting, in which bone marrow stromal cell (BMSC)-laden anisotropic hydrogel micropatterns were utilized for heterogeneous chondrogenic differentiation and physically gradient synthetic poly (ε-caprolactone) (PCL) to give mechanical strength. The results of their study rendered a new prospect for the regeneration and repair of cartilaginous tissue [30].

### 3.15. LncRNA-CRNDE Regulates BMSC Chondrogenic Differentiation to Promote Cartilage Repair in Osteoarthritis through SIRT1/SOX9

Shi et al. observed that the long noncoding RNA (lncRNA)-colorectal neoplasia differentially expressed gene (CRNDE) controlled BMSC chondrogenic differentiation to facilitate the repair of cartilaginous tissue in OA via SIRT1/SOX9 [7].

### 3.16. Manipulation of Histone Signals to Control Chondrocyte Roles or Manage Injuries of the Cartilaginous Tissue

Wan et al. affirmed that the utilization of small molecules and medications can manipulate histone signals to control chondrocyte roles or manage cartilage injuries and OA [31].

### 3.17. The Function of Histone Deacetylase 4 (HDAC4)-Controlled Chondrocyte Hypertrophy in the Start and Appearance of Age-Related OA

Dong et al. reported that HDAC4 expression controls the beginning and appearance of age-related OA by regulating chondrocyte hypertrophy. Therefore, the findings of this study might aid in the early diagnosis and management of age-related OA [32].

### 3.18. The Impact of Osmolarity and FK506 (Tacrolimus, Fujimycin) on Calcineurin Action, Cell Expansion, Extracellular Matrix Quality, and BMP- and TGF-β (Transforming Growth Factor Beta) Signaling

Jahr et al. discovered that physiological osmolarity facilitated terminal chondrogenic differentiation of progenitor cells via the sensitization of the TGF-β superfamily signaling at the type I receptor. The findings of this study could be of help to future cell-based cartilage repair approaches [33].

### 3.19. Mechanism by which OA Is Started and Advances in the Cartilage Extracellular Matrix (ECM)

Jaabar et al. showed the likelihood of imitating the unevenness actions of chondrocytes by employing enzymatic digestions of healthy cartilage, by means of the compound action of hyaluronidase and collagenase. This produced damage completely similar to that seen in advanced OA. Targeting the homeostatic equilibrium of chondrocyte metabolism via the regulation of enzymatic responses implicated in catabolic processes could be a potential curative treatment of OA [9].

### 3.20. Proteoglycan 4 (Prg4)/Lubricin Functions in a Protective Manner against OA

Takahata et al. reported that proteoglycan 4 (Prg4)/lubricin, which is in a specific manner expressed in the external area of AC and the synovial membrane, protects against OA, and controls the transcriptional regulation of Prg4 in chondrocytes of AC [10].

### 3.21. Chondroinductive/Chondroconductive Peptides and Their Functionalized Biomaterials for Cartilage Tissue Engineering

According to Zhu et al., peptides that are derived from and imitate the roles of chondroconductive cartilage ECM and chondroinductive growth factors, constitute a distinctive category of bioactive drugs for chondrogenic functionalization. Taking into account that they can be chemically synthesized, peptides exhibit superior reproducibility, more stable effectiveness, better modifiability, and producing efficacy compared to naturally derived biomaterials and recombinant growth factors. Therefore, peptides can be utilized for cartilage tissue engineering [12].

### 3.22. Reversal of Tissue Degradation Detected with Joint Distraction

According to Jansen et al., the reversal of tissue degradation found with joint distraction could be due to one or a compound of diverse processes, including partial unloading, fluctuation of the pressure of the synovial fluid, mechanical and biochemical modifications in subchondral osseous tissue, adhesion and chondrogenic dedication of articulation-derived MSCs, or a modification in the molecular ambience of the articulation [34].

### 3.23. α2-Macroglobulin (α2M) Is Important to Chondral Protection in Post-traumatic OA

Zhao et al. showed that α2MRS injection substantially lowered the levels of inflammatory factors, ameliorated gait, and demonstrated substantially inferior degradation of cartilaginous tissue than the groups that did not experience α2M-rich serum (α2MRS) injections. This study emphasized the chondroprotective impact of α2MRS, clarified its possible utilizations against cartilage degradation, and could render a basis for the clinical translation of α2MRS [35]. 

### 3.24. The Potential Role of Nano-Antioxidants

It has been reported by Khezri et al. that curcumin nanoformulations have several pleiotropic pharmacological effects. That is why curcumin, a natural compound, can be used in the treatment of osteoarthritis. The impact of curcumin and its nanoformulations on the differentiation of MSCs has been shown. Additionally, osteogenic differentiation of MSCs in the scaffolds has been demonstrated [36].

### 3.25. The Possible Role of Mitochondria Targeting in Tissue Repairing

It has been reported by Chodari et al. that adequate mitochondrial biogenesis is required for efficacious cell function and homeostasis, which depends on the control of ATP production and preservation of mitochondrial DNA (mtDNA). These phenomena are essential in the mechanisms of inflammation and aging, among others. Polyphenols have been deemed as the principal components of plants, fruits, and natural extracts with demonstrated long lasting therapeutic impact. These components control the intracellular pathways of mitochondrial biogenesis. The impact of several natural polyphenol compounds from various plant kingdoms on regulating signaling pathways of mitochondrial biogenesis make them potential options for the management of OA and defects of AC [37].

A study suggested that intercellular transfer of healthy mitochondria to chondrocytes could represent a new, acellular approach to increase mitochondrial content and function in cartilage [38].

### 3.26. The Use of Infrapatellar Fat Pad-Derived Mesenchymal Stem Cells in Articular Cartilage Regeneration

According to Vehadi et al., the origin of MSCs is a relevant factor to be taken into account. The infrapatellar fat pad (IPFP) is a rich source of MSCs. Additionally, it has been demonstrated that these cells have many pros over other tissues in terms of ease of isolation, expansion, and chondrogenic differentiation. Therefore, it is why IPFP-derived MSCs are already being used in orthopedic surgery for the treatment of OA and other injuries of AC [39]. Table 1 summarizes the main data from the literature on molecular processes of cartilage repair and their potential clinical applications.

## 4. Conclusions

Hyaluronic acid triggers the RASL11B gene to augment the chondrogenic differentiation of hAMSCs through the stimulation of Sox9 and ERK/Smad signaling, thus rendering a new approach for the repair of cartilage defects by hyaluronic acid-based stem cell therapy. ADSCs have shown to be anti-OA efficacious and safe, ADSCs-CM facilitated the rapid growth in the number of chondrocytes and substantially re-established the IL-1β-induced abnormal expressions of molecular markers IL-6, aggrecan, MMP3, MMP13, collagen II, collagen X, ADAMTS5, ADAMTS9, SOX6, and SOX9 in chondrocytes. Compared to MSCs, MSC-derived exosomes have many pros such as non-immunogenicity, easy access, easy storage, and great stability under different circumstances. Exosomes could be deemed as an alternate approach for cell-based treatment in regenerative medicine. The 3N-cadherin can facilitate chondrogenic differentiation of BMSCs by restraining the Wnt signaling pathway. Nanotechnology can be valuable for the diagnosis, follow-up, and treatment of OA. HPT has substantial protective and anti-inflammatory impact on chondrocytes via the AMPK signaling pathway, efficaciously averting cartilage degeneration. lncRNA-CRNDE controls BMSC chondrogenic differentiation to facilitate cartilage repair in OA via SIRT1/SOX9. HDAC4 expression controls the start and appearance of age-related OA by regulating chondrocyte hypertrophy. Prg4/lubricin functions in a protective mode against OA. It has been shown that α2-macroglobulin (α2M) protects against post-traumatic OA.

The suggestion of this article for future work is that we must continue to continuously and relentlessly investigate the intimate mechanisms of articular cartilage degeneration with aging in order to find therapeutic solutions to slow or prevent it. Additionally, efforts should be made to try to improve the healing capacity of chondrocytes with various therapeutic strategies, such as those discussed in this article.

## Figures and Tables

**Figure 1 ijms-23-14272-f001:**
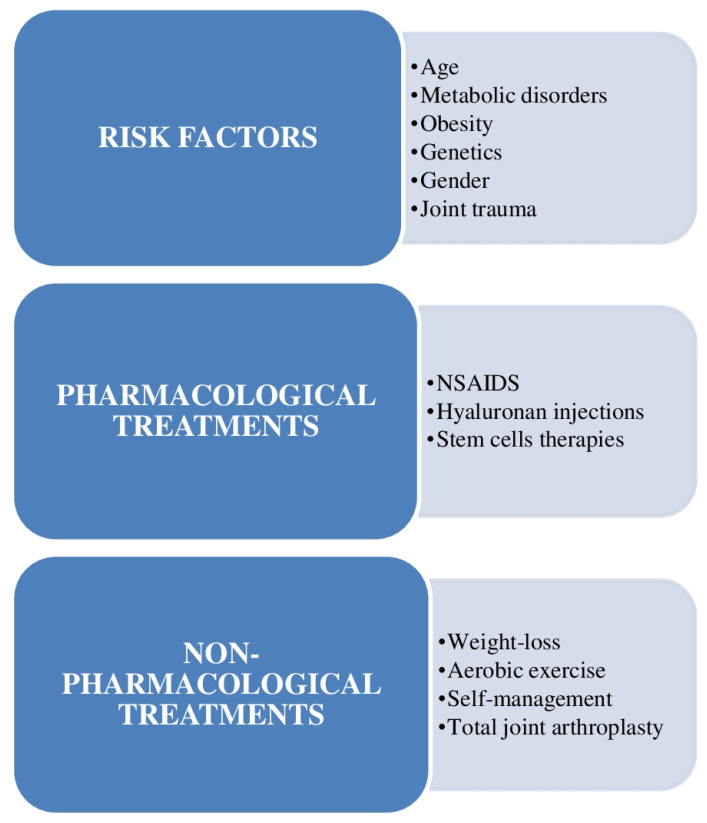
Outline of the risk circumstances for the appearance of osteoarthritis (OA) and the contemporary pharmacological and nonpharmacological approaches for its management, emphasizing the scarcity of efficacious methods and the chance for invention in this field. NSAIDS = nonsteroidal anti-inflammatory drugs.

**Figure 2 ijms-23-14272-f002:**
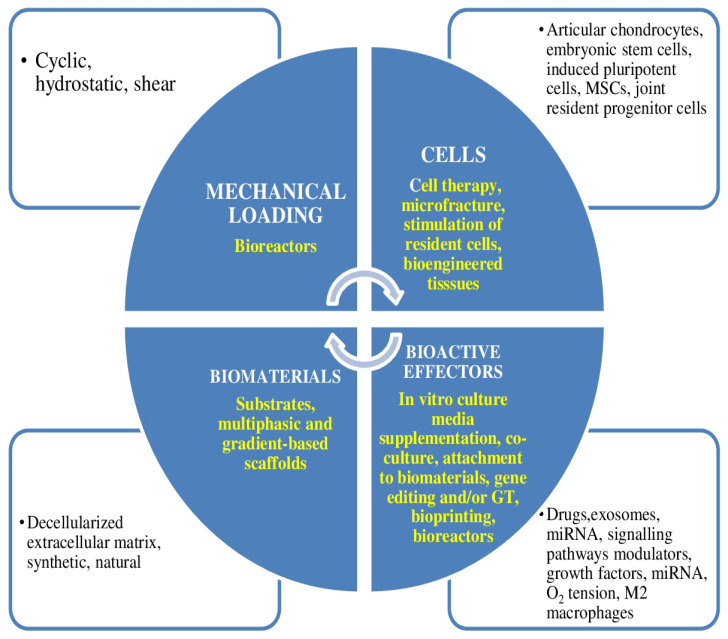
Elements implicated in cellular therapy and tissue engineering for the repair of cartilaginous tissue. Therapies for cartilage repair need any compound of cells, biomaterials, mechanical loading, and/or bioactive effectors. In yellow, the four main ways in which they can be used for cartilage tissue engineering are shown. MSCs = mesenchymal stem cells.

**Figure 3 ijms-23-14272-f003:**
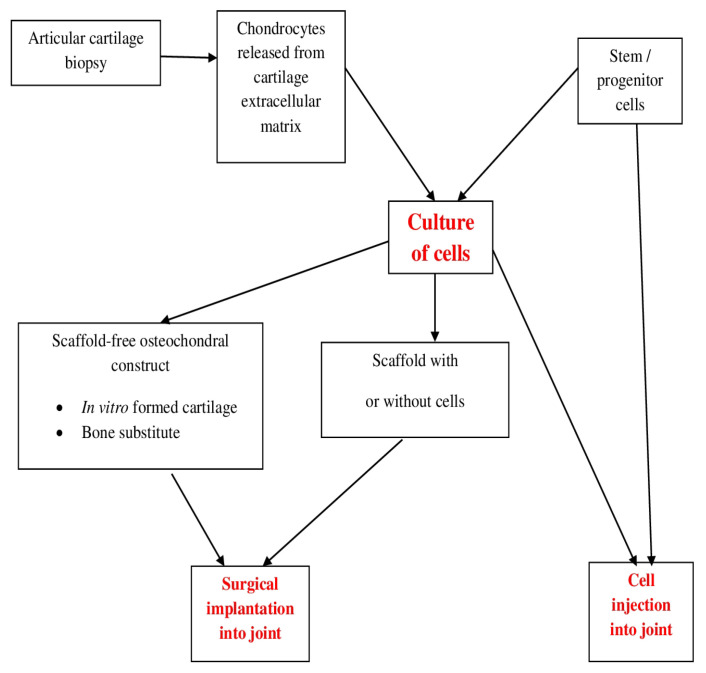
Outline of approaches for the repair of cartilaginous tissue. All methods exhibited can be improved with bioactive effectors or mechanical loading.

**Figure 4 ijms-23-14272-f004:**
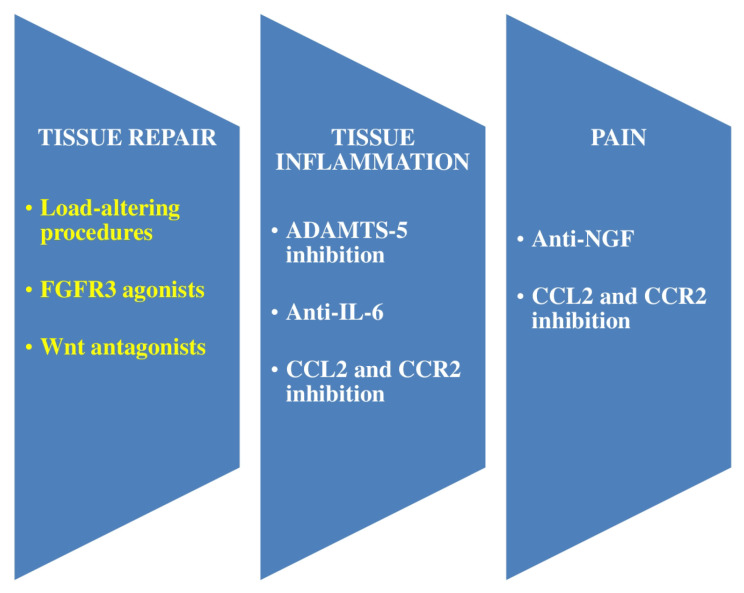
Main emergent therapeutic objectives in osteoarthritis (OA) are shown. Pathological objectives mainly gather into those facilitating repair, those neutralizing pain, and those preventing tissue inflammation (resulting in turn in degeneration). There is a mutual correlation between inflammatory and repair processes in the OA articulation, both of which impact pain. Objectives that demonstrate effectiveness in murine experiments and for which therapeutic approaches are being studied in clinical trials are shown. White color indicates promotion, and yellow color represents suppression. Peripheral pain emerges from joint pathology and may prevent tissue swelling and facilitate tissue repair by suppressing mechanical overload of the articulation. FGFR = fibroblast growth factor receptor. IL = interleukin. ADAMTS = a disintegrin and metalloproteinase with thrombospondin motif. CCL = C-C motif chemokine. CCR = C-C chemokine receptor. NGF = nerve growth factor.

**Figure 5 ijms-23-14272-f005:**
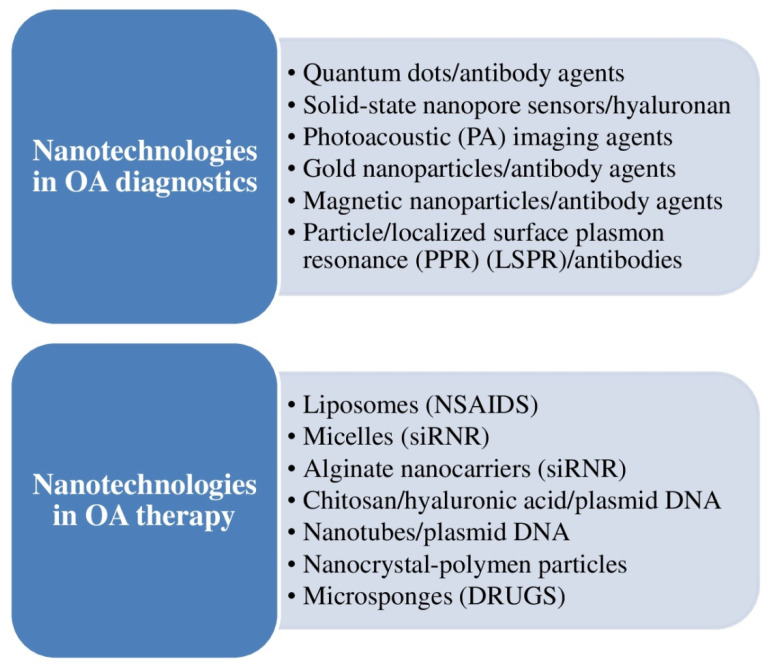
Outline of nanotechnology-based uses in the diagnosis and treatment of osteoarthritis (OA).

**Table 1 ijms-23-14272-t001:** Main data from the literature on molecular processes of cartilage repair and their potential clinical applications.

Authors Reference	Year	Methods and Results	Conclusicons
Vincent [14]	2020	Genome-wide studies pointed to defects in repair pathways, which agreed well with contemporary potential utilizing growth factor treatment or Wnt pathway antagonism.	Nerve growth factor appeared as a good objective in OA pain in phase 2–3 trials.
Salerno et al. [17]	2020	In a try at developing functional markers of MSCs potency, Salerno et al. recognized loss of mRNA expression for MMP13 as correlating with loss of chondrogenic potential of MSCs and continued secretion of elevated levels of TIMP1 protein as correlating with the maintenance of trophic repair ability.	Early passage MMP13^+^, TIMP1-secreting^high^ MSCs should be utilized for autologous OA treatment, while later passage MMP13^−^, TIMP1-secreting^high^ MSCs could be used for allogeneic OA treatment.
Luo et al. [15]	2020	Microarray analysis demonstrated that RASL11B played an essential role in the process of hyaluronic acid-mediated chondrogenesis of hAMSCs.	Hyaluronic acid triggered the RASL11B gene to potentiate the chondrogenic differentiation of hAMSCs through the activation of Sox9 and ERK/Smad signaling.
Ross et al. [24]	2020	These authors carried out miRNA and mRNA sequencing to establish the temporal and dynamic reactions of genes to specific inflammatory cytokines as well as miRNAs that are differentially expressed (DE) in reaction to both cytokines or exclusively to IL-1β or TNF-α.	This study utilized an integrative approach to establish the miRNA interactome controlling the reaction to inflammatory cytokines.
Lu et al. [5]	2020	This study showed that SIRT1 could facilitate BMP2-induced chondrogenic differentiation of MSCs and diminish the apoptosis and decomposition of ECM under oxidative stress.	SIRT1 played an important coordination role in BMP2-induced chondrogenic differentiation of stem cells and cartilage maintenance under oxidative stress.
Zhang et al. [18]	2020	To facilitate comprehension of the molecular regulation of chondrogenesis differentiation in MSCs, these authors compared the modifications in miRNAs during in vitro chondrogenesis process of hBMSCs. MiR-199b-5p was upregulated substantially during this process.	miR-199b-5p was the positive regulator to modulate chondrogenic differentiation of C3H10T1/2 cells by aiming JAG1.
Mohammadinejad et al. [11]		This review article discussed the possible utilization of nanotechnological approaches for the diagnosis, follow-up, and treatment of OA and analyzed how nanotechnology was being integrated quickly into regenerative medicine for OA.	Nanotechnology platforms might be combined with cell, gene, and biological treatment for the discovery of a new generation of future OA drugs.
Travascio et al. [3]	2020	This study analyzed the functions of solute size and tissue composition on molecular diffusion in knee AC. Diffusivity was influenced by molecular size, with the magnitude of the diffusion coefficients diminishing as the Stokes radius of the probe augmented.	This study rendered new data on the processes of diffusion in AC. The results of the study can be used to further examine OA and to design approaches for AC restoration or replacement.
Du et al. [26]	2020	These authors studied the expression of EZH2, an H3K27me3 transferase, in human OA cartilages and its functions in controlling OA pathogenesis.	Utilizing histological analysis and RNA sequencing (RNA-Seq), it was noticed that EZH2 was highly expressed in both mice and human OA cartilage.
Wu et al. [27]	2021	This study evaluated the potential chondroprotective effects of hesperetin (HPT) against the TNF-α-induced inflammatory reaction of chondrocytes and related processes, and elucidated the influence of HPT on OA induced by anterior cruciate ligament transection (ACLT).	HPT had significant protective and anti-inflammatory impact on chondrocytes via the AMPK signaling pathway, efficaciously averting cartilage degeneration.
Casari et al. [19]		These authors compared adipose tissue (AT) processed by centrifugation (C-AT) to microfragmentation (MF-AT). They evaluated the impact of synovial fluid (SF) alone on both MF-AT and isolated AT-MSC to better comprehend their cartilage repair processes.	MF had a positive impact on the maintenance of AT histology and might initiate the expression of trophic factors that ameliorate tissue repair by processed AT.
Keppie et al. [1]	2021	This study described three pro-regenerative factors, fibroblast growth factor 2 (FGF2), connective tissue growth factor, bound to transforming growth factor-beta (CTGF-TGFβ), and hepatoma-derived growth factor (HDGF), that were quickly liberated from the pericellular matrix (PCM) of AC upon mechanical injury.	Loss of aggrecan in late-stage OA averted growth factor liberation and likely contributed to illness progression.
Zhong et al. [6]	2021	These authors explored the possible function and process of circadian gene clock in OA pathology.	Clock knockdown attenuated ROS-mediated senescence of chondrocytes via reestablishing autophagic flux in noncircadian mode.
Liu et al. [4]	2021	This experimental study aimed to determine the proliferative reactions of chondrocytes to negative pressure and explored the possible molecular processes.	HMGB2 facilitated chondrocyte proliferation under negative pressure via the phosphorylation of AKT.
van Schaik et al. [28]	2021	In this study, the authors used an ex vivo murine osteochondral repair model utilizing human infrapatellar fat pad (IPFP) progenitor cells.	These authors showed the regenerative potential of OA human IPFP-derived progenitors in mouse femurs.
Yang et al. [29]	2021	These authors reviewed the role of RGD stated Arginine–Glycine–Aspartic acid (Arg–Gly–Asp, RGD) peptide family in osseous and cartilaginous tissues engineering.	RGD peptide via integrin was introduced in the discipline of osseous and cartilaginous tissues engineering.
Sun et al. [30]	2021	In this study, Sun et al. constructed an anisotropic gradient-structured cartilage scaffold by three-dimensional (3D) bioprinting, in which bone marrow stromal cell (BMSC)-laden anisotropic hydrogel micropatterns were utilized for heterogeneous chondrogenic differentiation and physically gradient synthetic PCL to achieve mechanical strength.	The results of this study rendered a theoretical basis for using 3D bioprinting gradient-structured constructs for anisotropic cartilage regeneration.
Shi et al. [7]	2021	This study explored the function of long noncoding RNA (lncRNA)-colorectal neoplasia differentially expressed gene (CRNDE) in the chondrogenic differentiation of BMSCs and the intrinsic molecular process, seeking to create a new therapeutic approach for OA.	Overexpression of lncRNA-CRNDE augmented the binding capacity of SOX9 and col2α1 promoter, which was reversed by the concomitant transfection of CRNDE overexpression (pcDNA-CRNDE) and SIRT1 small interfering RNA (si-SIRT1).
Wan et al. [31]	2021	These authors stated that comprehending the molecular processes and impact of histone modification enzymes in cartilage development, homeostasis, and pathology will render essential and precise prospects to understand the biological behavior of chondrocytes during skeletal development and the pathogenesis of several cartilage-related illnesses.	These authors emphasized the importance of using small molecules and drugs to manipulate histone signals to treat cartilage lesions and OA.
Ren et al. [20]	2021	In this study, Ren et al. stated that the stemness and differentiation characteristics of BMSCs in 3D culture were of great importance for stem cell treatment and cartilage tissue engineering repair. Furthermore, because of their mechanical sensitivity, scaffold materials played relevant functions in various cell behaviors in 3D culture.	This research could be valuable for designing biomaterials for BMSCs’ delivery in vivo, as well as for developing cartilage repair drug delivery programs based on molecular processes.
Yan et al. [2]	2021	This study showed that the DNA supramolecular hydrogel can improve formation of quality cartilage, diminish bone spurs, and normalize subchondral osseous tissue in OA.	DNA supramolecular hydrogel could be a good cell delivery system for MSCs treatment.
Zhu et al. [12]	2021	These authors summarized the contemporary data in the designs of the chondroinductive/chondroconductive peptides, their intrinsic molecular processes, and their functionalized biomaterials for cartilage tissue engineering.	Peptides constitute a remarkable group of bioactive drugs for chondrogenic functionalization.
Khezari et al. [36]	2021	This article has reviewed the pharmacological effects of curcumin nanoformulations.	Nanocurcumin may benefit the osteogenic differentiation of MSCs in the scaffolds.
Chodari et al. [37]	2021	These authors reviewed effects of different natural polyphenol compounds from various plant kingdoms on modulating signaling pathways of mitochondrial biogenesis.	The use of polyphenol compounds is a promising alternative for the treatment of OA.
Vehadi et al. [39]	2021	These authors stated that the infrapatellar fat pad IPFP is a rich source of MSCs.	IPFP-derived MSCs are promising in the treatment of articular cartilage damage
Thomas et al. [38]	2022	This study suggested that intercellular transfer of healthy mitochondria to chondrocytes could represent a new, acellular approach to increase mitochondrial content and function in cartilage.	The intercellular transfer of healthy mitochondria to chondrocytes is a promising alternative for the treatment of OA.
Yan et al. [8]	2022	This study assessed the anti-OA effectiveness of adipose-derived stem cells (ADSCs) and explored the intrinsic mechanism of action.	The anti-OA efficacy, safety, and a paracrine-based mechanism of ADSCs was shown.
Zhao et al. [35]	2022	In this study, Zhao et al. assessed the protective role of of alpha2MRS against post-surgery cartilage degradation.	This study highlighted the chondroprotective impact of alpha2MRS and rendered a basis for the clinical translation of alpha2MRS.
Norouzi-Barough et al. [21]	2022	MSC-derived exosomes can imitate the biological activity of MSCs by horizontal transfer of many molecules including mRNAs, miRNAs, proteins, and lipids to the local microenvironment and recipient cells.	Exosomes could be deemed as an alternative approach for cell-based treatments in regenerative medicine.
Dong et al. [32]	2022	In transgenic and control mice, these authors tried to define the function of histone deacetylase 4 (HDAC4)-controlled chondrocyte hypertrophy in the start and appearance of age-related OA.	HDAC4 expression controlled the start and appearance of age-related OA by regulating chondrocyte hypertrophy.
Jahr et al. [33]	2022	This investigation aimed at clarifying the function of osmolarity in chondrogenic differentiation.	Physiological osmolarity improved terminal chondrogenic differentiation of progenitor cells.
Feng et al. [22]	2022	This study compared and analyzed the impact of N-cadherin on chondrogenic differentiation of bone marrow-derived mesenchymal stem cells (BMSCs) and explored the related mechanism.	N-cadherin can facilitate chondrogenic differentiation of BMSCs by restraining the Wnt signaling pathway.
Zhang et al. [23]	2022	This study described a tetracycline (tet)-controlled bone morphogenetic protein-4 (BMP-4) expressing the iPSC (iPSC-Tet/BMP-4) line in which transcriptional activation of BMP-4 was associated with improved chondrogenesis.	The culture system employed could be a valuable instrument for further research of the mechanism of BMP-4 in controlling iPSC differentiation toward the chondrogenic lineage.
Galla et al. [16]	2022	In this study, Galla et al. confirmed the effectiveness of a new high molecular weight hyaluronic acid of plant origin (called GreenIuronic^®^) in preserving the homeostasis of the joints and averting the detrimental mechanisms of OA.	The new form of HA reported in this study appeared to be well-absorbed and distributed to chondrocytes.
Li et al. [25]	2022	These authors provided a systematic review of the latest progress of extracellular vesicles (EVs) for regenerative applications.	The bioactive EV-based nanotherapeutics have opened new horizons for clinicians, making feasible strong tools and treatment for regenerative medicine.
Jaabar et al. [9]	2022	In this study, Jaabar et al. used X-ray photoelectron spectroscopy (XPS), and created a new method that rendered the molecular composition of the ECM.	Aiming the homeostatic equilibrium of chondrocyte metabolism via the control of enzymatic reactions implicated in catabolic processes could be a new therapeutic approach against OA.
Takahata et al. [10]	2022	This review article showed that proteoglycan 4 (Prg4)/lubricin protects against OA.	Molecular targeting of Prg4 and GDF5 could be a potential treatment of OA.
Zelinka et al. [13]	2022	These authors stated that cell-based techniques have entered clinical practice in orthopedic surgery and that some tissue engineering strategies to repair AC are about to be clinically employed.	Comprehending the intrinsic mechanisms of action of cell therapy and tissue engineering techniques will be helpful in the future.
Jansen et al. [34]	2022	Several mechanisms are implicated in the effect of joint distraction.	Joint distraction is a joint-preserving management for end-stage OA.

OA = osteoarthritis; FOXM1 = forkhead transcription factor M1; CXCL12 = C-X-C motif chemokine ligand 12; MSCs = mesenchymal stem cells; MMP = metalloproteinase; TIMP1 = TIMP metallopeptidase inhibitor 1; Col2α1= collagen type 2 alpha 1; SOX9 = SRY-box transcription factor 9; hAMSCs = human amniotic mesenchymal stromal cells; TGF-β 3 = transforming growth factor β3; RASL11B = ras-like protein family member 11B; ERK = extracellular signal-regulated protein kinase; miRNA = microribonucleic acid (RNA); mRNA = messenger RNA; IL-1β = interleukin beta; TNF-α = tumor necrosis factor alpha; SIRT1 = silent mating type information regulator 2 homolog-1; BMP-2 = bone morphogenetic protein-2; ECM = extracellular matrix; hBMSCs = human bone-marrow mesenchymal stem cells; JAG1 = jagged canonical notch ligand 1; AC = articular cartilage; EZH2 = enhancer of zeste 2 polycomb repressive complex 2 subunit; H3K27me = trimethylation of histone H3 lysine 27; HMGB2 = high mobility group box 2; iPSC = induced pluripotent stem cells; GDF5 = growth differentiation factor 5; RNA-seq (RNA sequencing); AMPK = adenosine monophosphate (AMP)-activated protein kinase; qRT-PCR = quantitative real time-polymerase chain reaction; ROS = reactive oxygen species; HMGB2 = high mobility group box 2; HIF1 alpha/FAK = hypoxia-inducible factor 1 alpha/focal adhesion kinase; pcDNA = plasmid cloning DNA (deoxyribonucleic acid); Alpha2MRs = alpha 2-macroglobulin receptor; TGF-β3 = transforming growth factor-β3.

## Data Availability

Not applicable.

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
