# Peer review of "Molecular Mechanisms of Cartilage Repair and Their Possible Clinical Uses: A Review of Recent Developments"

_ijms, 2022, doi:10.3390/ijms232214272_

Round 1

Reviewer 1 Report

Osteoarthritis afflicts millions of individuals across the world resulting in impaired quality of life and increased health costs. Before starting treatment, the patient wants to receive from the doctor a guarantee of a successful end result of the treatment, and at the same time, the price of the treatment must remain within adequate limits. Such mutually competing trends predetermine the difficult choice of the optimal tactics and treatment strategy for each specific case. The article presented for consideration will help doctors find optimal solutions to similar practical problems.

The cartilage tissue belongs to the family of bradytrophic connective tissues. Due to its macro- and microstructural composition and its highly organized structure, cartilage tissue poses a great challenge for doctors who want to repair and regenerate this highly specific tissue. The  problem is that although cartilage tissue has a basic regenerative potential, it actually contains additional progenitor cells, mesenchymal stem cells, which are the basic prerequisite for adequate tissue regenration. However, the repair tissue is not capable of withstanding physiological stress on the cartilage tissue in the joints, resulting in joint degeneration, the impairment of joint use, dysfunction, deformation, pain etc.

Modern treatment approaches such as endoprosthetics, multiple drilling, microfractures, autologous chondrocyte implantation, and joint replacement and others are associated with many side effects. In addition, there are still many limitations in creating an adequate cellular microenvironment – the scaffolds used must be biocompatible, biodegradable, non-toxic, and have the required mechanical properties, and the materials used can be expensive and have not yet been fully tested.

Results

In view of the foregoing, it can be argued with confidence that the topic of the presented article is very relevant. The manuscript itself is a qualitatively and very diligently executed review article. The results obtained in this study are of scientific and practical value. The article will be of interest to specialists in the relevant industry, and I can also predict that it will be highly cited. In my opinion, this article deserves to be published as presented.

Author Response

REVIEWER-1

Osteoarthritis afflicts millions of individuals across the world resulting in impaired quality of life and increased health costs. Before starting treatment, the patient wants to receive from the doctor a guarantee of a successful end result of the treatment, and at the same time, the price of the treatment must remain within adequate limits. Such mutually competing trends predetermine the difficult choice of the optimal tactics and treatment strategy for each specific case. The article presented for consideration will help doctors find optimal solutions to similar practical problems.

The cartilage tissue belongs to the family of bradytrophic connective tissues. Due to its macro- and microstructural composition and its highly organized structure, cartilage tissue poses a great challenge for doctors who want to repair and regenerate this highly specific tissue. The problem is that although cartilage tissue has a basic regenerative potential, it actually contains additional progenitor cells, mesenchymal stem cells, which are the basic prerequisite for adequate tissue regenration. However, the repair tissue is not capable of withstanding physiological stress on the cartilage tissue in the joints, resulting in joint degeneration, the impairment of joint use, dysfunction, deformation, pain etc.

Modern treatment approaches such as endoprosthetics, multiple drilling, microfractures, autologous chondrocyte implantation, and joint replacement and others are associated with many side effects. In addition, there are still many limitations in creating an adequate cellular microenvironment – the scaffolds used must be biocompatible, biodegradable, non-toxic, and have the required mechanical properties, and the materials used can be expensive and have not yet been fully tested.

Results

In view of the foregoing, it can be argued with confidence that the topic of the presented article is very relevant. The manuscript itself is a qualitatively and very diligently executed review article. The results obtained in this study are of scientific and practical value. The article will be of interest to specialists in the relevant industry, and I can also predict that it will be highly cited. In my opinion, this article deserves to be published as presented.

AUTHOR: I do really appreciate the positive evaluation made by the Reviewer. Therefore, no changes have been made in the initial manuscript.

Reviewer 2 Report

The manuscript was well designed.
Also, the paper was nicely written and the authors reviewed suitable references to corroborate their hypothesis.
There some concerns needs to be improved.
1.It is suggested to add new research studies.
2.Authors should add more details for dose and period of treatment with references.
3.what is the suggestion of this study for future works?
4.Please discuss about the using of nanoantioxidants.
5.It will be useful if authors discuss the possible role of mitochondria targeting in tissue repairing.
6.Please add these references for your discussion part of manuscript and bold your study novelty :
-DOI: 10.3390/ijms22179215
-DOI: 10.1155/2021/4946711
-DOI: 10.1155/2021/1520052

Author Response

REVIEWER-2

The manuscript was well designed. Also, the paper was nicely written and the authors reviewed suitable references to corroborate their hypothesis.

AUTHOR: I do really appreciate the positive evaluation of my article made by the Reviewer.

There some concerns needs to be improved.

AUTHOR: Below and in the revised manuscript (REVISION-1) you can see IN RED the changes made following all your interesting suggestions:

  1. It is suggested to add new research studies.

AUTHOR: I have added the 3 important articles mentioned by the Reviewer in point 6.: DOI: 10.3390/ijms22179215 (Vahedi et al  - 2021); DOI: 10.1155/2021/4946711 (Chodari et al – 2021); DOI: 10.1155/2021/1520052 (Khezri et al  - 2021).

  1. Authors should add more details for dose and period of treatment with references.

AUTHOR: With regard to the Reviewer's suggestion to add more details on the dosage and treatment time of the various authors, I confess that I have not made any changes to the initial article given the great difficulty of expressing in a few words the great complexity of the studies analyzed in terms of these parameters. I am sure that interested readers will be able to find such information in the articles reviewed in this paper.  I hope this does not prevent the article from being accepted for publication.

  1. What is the suggestion of this study for future works?

AUTHOR: I have included the following paragraph in the “Conclusions”

The suggestion of this article for future work is that we must continue to continuously and relentlessly investigate the intimate mechanisms of articular cartilage degeneration with aging in order to find therapeutic solutions to slow or prevent it. Also, that efforts should be made to try to improve the healing capacity of chondrocytes with various therapeutic strategies, such as those discussed in this article.

  1. Please discuss about the using of nanoantioxidants.

AUTHOR: I have added a new subheading in SECTION 3 commenting the important article of Khezri et al, and included the reference in the Table:

3.24. The potential role of nanoantioxidants

It has been reported by Khezri et al that curcumin nanoformulations have several pleiotropic pharmacological effects. That is why curcumin, a natural compound, can be used in the treatment of osteoarthritis. The impact of curcumin and its nanoformulations on the differentiation of MSCs has been shown. Besides, osteogenic differentiation of MSCs in the scaffolds has been demonstrated [36].

  1. It will be useful if authors discuss the possible role of mitochondria targeting in tissue repairing.

AUTHOR: I have added a new subheading in SECTION 3 commenting the important article of Chodari et al, and a study of Thomas et al (both have been included the reference in the Table):

3.25. The possible role of mitochondria targeting in tissue repairing.

It has been reported by Chodari et al that adequate mitochondrial biogenesis is required for efficacious cell function and homeostasis, which depends on the control of ATP production and preservation of mitochondrial DNA (mtDNA). These phenomena are essential in the mechanisms of inflammation and aging, among others. Polyphenols have been deemed as the principal components of plants, fruits, and natural extracts with demonstrated long lasting therapeutic impact. These components control the intracellular pathways of mitochondrial biogenesis. The impact of several natural polyphenol compounds from various plant kingdoms on regulating signaling pathways of mitochondrial biogenesis make them potential options for the management of OA and defects of AC [37].

A study suggested that intercellular transfer of healthy mitochondria to chondrocytes could represent a new, acellular approach to increase mitochondrial content and function in cartilage [38].

NEW REFERENCE

  1. Thomas, M. A.; Fahey, M. J.; Pugliese, B. R.; Irwin, R. M.; Antonyak, M. A.; Delco, M. L. Human mesenchymal stromal cells release functional mitochondria in extracellular vesicles. Front Bioen Biotechnol 2022, 10, 870193.

  1. Please add these references for your discussion part of manuscript and bold your study novelty:

AUTHOR: As I have already commented on the articles of Chodari et al, and Khezri et al, I have added here a new subheading in SECTION 3 mentioning the important article of Vahedi et al, and included it in the Table:

3.26. The use of infrapatellar fat pad-derived mesenchymal stem cells in articular cartilage regeneration

According to Vehadi et al, the origin of MSCs is relevant factor to be taken into account. The infrapatellar fat pad (IPFP) is a rich source of MSCs. Besides, it has been demonstrated that these cells have many pros over other tissues in terms of ease of isolation, expansion, and chondrogenic differentiation. Therefore, it is why IPFP-derived MSCs are already been used in Orthopedic Surgery for the treatment of OA and other injuries of AC [39].